# In Vitro Toxicity of a DEHP and Cadmium Mixture on Sheep Cumulus–Oocyte Complexes

**DOI:** 10.3390/ijms26010005

**Published:** 2024-12-24

**Authors:** Antonella Mastrorocco, Letizia Temerario, Valeria Vurchio, Susanna Cotecchia, Nicola Antonio Martino, Maria Elena Dell’Aquila

**Affiliations:** Department of Biosciences, Biotechnology and Environment, University of Bari Aldo Moro, 70125 Bari, Italy; letizia.temerario@uniba.it (L.T.); valeria.vurchio@unimi.it (V.V.); susanna.cotecchia@uniba.it (S.C.); nicola.martino@uniba.it (N.A.M.); mariaelena.dellaquila@uniba.it (M.E.D.)

**Keywords:** sheep oocyte, cumulus cells, DEHP, cadmium, chemical mixture, in vitro maturation, mitochondria membrane potential, reactive oxygen species

## Abstract

Di-(2-ethylhexyl) phthalate (DEHP) and Cadmium (Cd) affect female reproduction. To date, toxicological research has focused on the effects of individual contaminants, whereas living beings are exposed to mixtures. This study analyzed the effects of a DEHP/Cd mixture on nuclear and cytoplasmic maturation of sheep cumulus–oocyte complexes (COCs) compared with single compounds. COCs recovered from slaughterhouses-derived sheep ovaries were in vitro exposed to 0.5 μM DEHP, 0.1 μM Cd, or DEHP/Cd mixture at the same concentrations during 24 h of in vitro maturation (IVM). After IVM, oocyte nuclear chromatin configuration was evaluated, and bioenergetic/oxidative parameters were assessed on expanded cumulus cells (CCs) and matured oocytes (chi-square test and one-way ANOVA; *p* < 0.05). Under examined conditions, oocyte nuclear maturation was never impaired. However, COC bioenergetics was affected with stronger effects for the mixture than single compounds. Indeed, the percentages of matured oocytes with healthy mitochondrial distribution patterns were reduced (*p* < 0.001 and *p* < 0.05 for mixture and single compounds, respectively). Oocyte mitochondrial membrane potential, intracellular ROS levels, and mitochondria/ROS co-localization were reduced, with the same significance level, in all contaminated conditions. CCs displayed increased ROS levels only upon mixture exposure (*p* < 0.001). In conclusion, in vitro exposure to the DEHP/Cd mixture affected COC quality in the sheep to a greater extent than separate compounds.

## 1. Introduction

Infertility is defined as the failure to achieve pregnancy after at least 12 months of regular unprotected sexual intercourse. It affects about 70 million people worldwide, and female factors account for about 37% of cases [1,2]. Infertility is caused by identifiable abnormalities or underlying diseases in 85% of infertile couples. The remaining 15% are reported as affected by “unexplained infertility” [1] that could be related to exposure to environmental chemicals that, according to Environmental Protection Agency (EPA), have been classified as Endocrine Disrupting Chemicals (EDCs): ‘‘Exogenous agents that interfere with the synthesis, secretion, transport, metabolism, binding action, or elimination of natural blood-borne hormones that are present in the body and are responsible for homeostasis, reproduction and developmental processes’’ [3]. Indeed, it is increasingly recognized that EDCs may be harmful to human and animal health, causing reproductive disorders [4,5,6,7,8] and other diseases [9,10,11].

EDC production has enormously increased during the last decades because EDCs are widely used in society. Indeed, they are ubiquitous and commonly found in natural settings as they are in a wide range of products, and many of them have a prominent role in the food chain since they can easily contaminate fish, meat, dairy, and poultry products [12]. So, humans and animals are exposed to a broad range of EDCs, mainly through diet but also inhalation and dermal uptake. Phthalates, including diethylhexyl phthalate (DEHP), and heavy metals such as cadmium (Cd) are widely diffused EDCs reported as affecting the female reproductive system.

DEHP is one of the most used industrial chemicals known as a “plasticizer” to make plastic material harder, more flexible and more durable. It is present in a wide range of products, such as packaging for food and beverages, pharmaceutical products for personal care, children’s toys, and medical equipment [12]. After sunlight exposure or food package treatments, DEHP may leach out of the plastic material into the food chain [13,14,15], and it is able to enter the body by oral ingestion [16], inhalation (house dust [17,18,19]), or dermal contact (through using personal care products or handling medical devices [20,21]). DEHP has been detected in a wide variety of human specimens, such as serum [22,23], urine [24,25], semen [24,26], and breast milk [27]. DEHP has also been reported in animal biological samples such as in bovine milk [28] and in ovine [29] and swine [30] tissue. Recent studies have highlighted the presence of DEHP also in human follicular fluid and established a direct correlation between exposure to this compound and detrimental effects on ovaries [19,31,32,33,34,35,36]. These findings confirm previous studies performed in animal models, providing evidence that exposure to DEHP can impair folliculogenesis and oocyte maturation and adversely affect embryo development [7,8,37,38,39,40,41,42,43,44,45,46,47,48].

Additional dangerous pollutants include Cd, a toxic non-essential transition metal widely used in consumer products, including cigarettes, batteries, and jewels, and as a dye in plastics, ceramics, glassware goods, and paints [49]. So, according to the Agency for Toxic Substances and Disease Registry “https://www.atsdr.cdc.gov/csem/cadmium/Who-Is-at-Risk.html (accessed on 20 October 2024)”, specific professional categories or smokers are particularly exposed to Cd. Moreover, exposure to Cd takes place mainly through oral ingestion of contaminated water and food such as rice, potatoes, wheat, leafy salad vegetables, and other cereal crops, mollusks, crustaceans, oilseeds, and offal [50,51]. In areas with contaminated soils, house dust is also a potential route for Cd exposure [52,53]. Cd is known to have a long half-life, and once absorbed by the body, it is transported into the bloodstream via erythrocytes and albumin to be irreversibly accumulated in the liver, gut, and kidneys [54,55]. Cd can also accumulate in the ovaries and follicular fluid, adversely influencing the likelihood of pregnancy and live birth [56,57,58,59,60]. Cadmium acts by perturbating the process of folliculogenesis, causing developmental disorders of primordial follicles and increasing the number of atretic follicles [61]. Moreover, it impairs the oocyte quality and meiotic maturation rate, leading to a decrease in female fertility, as reported in in vivo [60,62,63] and in vitro [64,65,66,67,68] studies.

Given the occurrence of Cd and DEHP in food packaging materials, humans and animals have a high probability of being exposed to a mixture of these chemicals rather than a single contaminant. These two plastic additives are not covalently bound but simply mixed with plastic polymers, so inappropriate plastic use, disposal, and recycling may lead to their undesirable release from food packaging materials into food and feed [69]. To date, to the best of our knowledge, the joint effects (i.e., additive, synergistic, or attenuative) of Cd and DEHP on oocyte maturation have never been studied, and most currently performed toxicological research so far has focused on evaluating the effects of individual compounds. However, from an innovative perspective, it is crucial to underline the importance of moving from a research paradigm based on a single pollutant to assessing the potential risks of human and animal exposure to chemical mixtures defined by EFSA as “any combination of two or more chemicals that may contribute to effects on a receptor (human or environmental) regardless of source and spatial or temporal proximity” [70]. Evaluating the effects of combined exposure to multiple chemicals on reproductive cells is still a challenge since few in vivo reproductive toxicity studies have investigated mixtures of chemicals [71,72,73,74,75,76].

In this context, the aim of the present study was to evaluate, in the sheep model, the effects of in vitro exposure to a DEHP and Cd mixture on nuclear maturation and bioenergetic aspects of the developmental potential of cumulus–oocyte complexes (COCs) in comparison with controls and individual contaminants.

## 2. Results

In this study, the nuclear chromatin configuration at metaphase II with the first polar body extruded (MII + PB) was the endpoint aimed at evaluating the effects of contaminants on oocyte maturation, whereas bioenergetic parameters of expanded CCs and matured (MII + PB) oocytes were the endpoints aimed at evaluating the effects of contaminants on oocyte developmental potential.

### 2.1. Results

Preliminarily, DEHP toxicity was tested at the concentrations of 0.1 and 0.5 µM. Control oocytes were cultured in an IVM medium supplemented with/without a DEHP vehicle. DEHP concentrations were selected based on previous studies carried out on horse oocytes [38,44]. In this experimental part, denuded oocytes were analyzed for nuclear chromatin. Since no effects on nuclear maturation were found, we wanted to analyze any cytoplasmic toxicity effects of DEPH. Therefore, mature oocytes were analyzed for cytoplasmic bioenergetic parameters.

#### DEHP Altered the Bioenergetic/Oxidative Status of Ovine Oocytes Matured in Vitro

Cumulus–oocyte complexes (*n* = 301 COCs in three independent replicates) were exposed to 0.1 µM and 0.5 µM DEHP during IVM. As shown in Table 1, no difference was noted between the maturation rates of oocytes cultured in control conditions (CTRL) versus those cultured in the vehicle; thus, subsequent experiments were performed using only the vehicle as control. DEHP exposure during IVM did not alter the percentage of oocytes that reached the MII stage compared with control oocytes (Table 1). Similarly, no statistically significant difference was observed between treated and control oocytes in the percentage of oocytes remaining at the stage of GV and MI or showing abnormal nuclear chromatin configurations (Table 1).

In order to determine the effects of DEHP on oocyte bioenergetic/oxidative status, we used fluorescent labeling confocal microscopy in single matured ones obtained after IVM to analyze a set of energy/redox ooplasmic parameters, such as mitochondrial distribution pattern, mitochondrial membrane potential, intracellular ROS localization, and levels. As shown in Table 2, the mitochondrial distribution pattern did not vary in vehicle CTRL oocytes compared with CTRL. In the group of oocytes exposed to 0.5 μM DEHP, the percentage of oocytes showing healthy heterogeneous perinuclear and subplasmalemmal mitochondrial distribution patterns was significantly reduced (*p* < 0.05), and a corresponding increase in the percentage of oocytes showing small aggregates mitochondrial pattern (*p* < 0.05) was observed compared to control conditions. The increased trend in unhealthy mitochondria distribution patterns was also observed at 0.1µM DEHP, even if it was not statistically significant (*p* = 0.0537).

Figure 1A–C shows graphs representing MitoTracker Orange CMTM Ros and DCF fluorescence intensity and Overlap coefficient, which indicate oocyte mitochondrial membrane potential (ΔΨm), intracellular ROS levels, and mitochondria/ROS co-localization, respectively, in examined conditions. Values of these parameters did not vary between CTRL and vehicle CTRL groups. Thus, CTRL values were not presented, and data were converted in percentages with respect to those of vehicle CTRLs. It can be seen that in the group of mature oocytes exposed to DEHP 0.5 µM, ΔΨm was significantly reduced compared with oocytes cultured under vehicle CTRL conditions (*p* < 0.05, Figure 1A) and intracellular ROS levels, and the degree of mitochondria/ROS co-localization were significantly reduced, as shown by the reduced levels of DCF intensity and coefficient of overlap in DEHP-exposed oocytes compared with oocytes cultured under vehicle CTRL conditions (*p* < 0.001, Figure 1B,C). Exposure to 0.1 µM DEHP did not induce any difference compared with vehicle CTRL. Given the obtained results, the concentration of 0.5 µM DEHP was selected for subsequent experiments.

### 2.2. Results

Based on the observation of the DEHP dose-dependence curve, the concentration of 0.5 µM DEHP was selected for the mixture. Cadmium chloride (CdCl_2_) concentration at 0.1 µM was identified based on previous studies [64,77]. The effects of the DEHP/Cd mixture were tested and compared with those of 0.5µM DEHP and 0.1 µM CdCl_2_ individually. In this experimental part, cells from expanded cumulus oophorus, presumably derived from COCs containing mature oocytes, were collected and analyzed. Subsequently, the denuded oocytes were analyzed for nuclear chromatin, and only the mature ones were analyzed for cytoplasmic bioenergetic parameters.

#### 2.2.1. The DEHP/CD Mixture and Individual Compounds Similarly Affected Oocyte Bioenerget-Ic/Oxidative Status

A total of 484 oocytes were analyzed in five replicates for nuclear maturation. As shown in Table 3, any treatment (DEHP/Cd mixture, DEHP, and Cd) had no significant effect on the percentages of oocytes that were able to reach the MII + PB stage between each condition and vehicle CTRL and among conditions.

Mature (MII) oocytes from four out of five replicates were also assessed for their bioenergetic/oxidative status. Both the mixture of Cd and DEHP and each single compound altered the bioenergetic/oxidative status of mature oocytes compared to controls.

In detail (Table 4), the DEHP/Cd mixture, as well as both single compounds, significantly reduced the percentage of matured oocytes with healthy mitochondrial P/S distribution patterns. In the presence of the DEHP/Cd mixture, the damage was stronger than in single compounds, as can be seen from the very low percentage of mature oocytes with healthy mitochondrial distribution pattern (*p* < 0.001), a corresponding increase in oocytes showing small mitochondrial aggregates (*p* < 0.001) and appearance of some oocytes with abnormal pattern with large irregular mitochondria clusters (Table 4).

In addition, oocytes matured in the presence of a DEHP/Cd mixture, DEHP or Cd showed significantly lower ΔΨm (*p* < 0.01), intracellular ROS levels (*p* < 0.001), and degree of co-localization mitochondria/ROS (*p* < 0.001) compared to vehicle CTRL, whereas no differences were noticed among contaminated conditions (mix versus DEHP; mix vs. Cd; DEHP vs. Cd) (Figure 2).

The bioenergetic damage induced in oocytes by DEHP/Cd mixture, DEHP, or Cd, is also evident in the confocal microscopy images shown in Figure 3. Whereas CTRL oocytes display a healthy P/S mitochondrial distribution pattern (Figure 3(A2)), oocytes exposed to the contaminants show a homogeneous mitochondrial distribution pattern typical of cytoplasmic immature oocytes (Figure 3(B2,C2,D2)). In addition, the marked decrease in the fluorescence signal in the oocytes exposed to the contaminants (Figure 3B–D) indicates the reduction in the bioenergetic/oxidative parameters compared to CTRL (Figure 3A).

#### 2.2.2. DEHP/CD Mixture Altered the Bioenergetic/Oxidative Status of Cumulus Cells

In three out of the five described replicates, we isolated and analyzed CCs from in vitro cultured COCs. Cumulus cells from 68, 64, 74, and 57 COCs cultured under CTRL, DEHP/Cd mixture, 0.5µM DEHP, and 0.1µM Cd, respectively, were assessed for bioenergetic parameters. Per each condition, 10 fields of about 20 cells each were observed for a total of approximately 200 CCs per condition. The diagrams of the analyzed parameters are shown in Figure 4. The experimental data show that intracellular ROS levels stained with DCDHFDA were significantly increased in the CCs of ovine oocytes in vitro matured in the presence of DEHP/Cd mixture compared to those matured under the other conditions (Figure 4B; *p* < 0.001), indicating oxidative stress (OS). Figure 5 shows representative photomicrographs of control CCs (A) and CCs exposed to DEHP/Cd mixture (B), DEHP (C), and Cd (D). Increased intracellular ROS levels in CCs exposed to the mixture (B2) can be observed.

## 3. Discussion

The novelty of this study is to analyze the effects of exposing COCs to a DEHP/Cd mixture during IVM on both COC somatic and germinal compartments (cumulus cells and the oocyte, respectively) and to compare them with those of each individual compound. It is well known that DEHP and Cd are included in the list of EDCs, which are exogenous compounds that may interfere with hormonal signals influencing reproductive functions [6]. DEHP and Cd have similar sources and exposure routes for both humans and animals that can often be exposed to their mixtures. Extensive research has revealed the effects of individual pollutants, while a significant gap persists in understanding the combined effects of their mixtures.

In this study, the sheep model has been used, which, besides being economically important worldwide for animal productions, is also useful for studying the potential effects of chemicals on oocyte maturation [78,79] and is a relevant translational animal model for human reproductive medicine [78,79,80].

We preliminarily analyzed the effects of two DEHP concentrations to identify an effective concentration on COCs of the animal model species used in this study. We found that 0.1 and 0.5 µM DEHP did not compromise oocyte nuclear maturation rates. This result is in line with data from studies performed in pigs [81] and equine oocytes exposed to DEHP [44]. Despite a lack of effect on oocyte nuclear maturation, our findings indicated that DEHP exposure, at the highest tested concentration (0.5µM), was associated with oocyte mitochondrial dysfunction, as demonstrated by a significant reduction in matured oocytes with healthy perinuclear/subplasmalemmal mitochondrial distribution patterns and decreased quantitative mitochondrial parameters. These results are in line with previous studies in mouse oocytes [82].

After selecting 0.5 µM as the toxic concentration of DEHP, our aim was to test the effects of the DEHP/Cd mixture on sheep COCs compared with those of each individual compound. Our findings indicated that exposing COCs to the mixture versus a single contaminant did not alter in vitro oocyte nuclear maturation but significantly impaired cytoplasmic maturation demonstrated by oocyte inability to properly direct mitochondria migration at perinuclear and subplasmalemmal cytoplasmic compartments. These data confirm previous observations on the effects of Cd [64] and DEHP [38], but the mixture has had stronger effects than individual compounds. Examined contamination conditions also induced mitochondrial dysfunctions, such as decreased membrane potential, ROS production, and mitochondria/ROS co-localization, with no differences among groups. Moreover, interestingly, when the effect of the contaminants was tested on the CCs derived from COCs cultured in vitro, we found that the DEHP/Cd mixture, but not each single contaminant, significantly increased ROS levels.

In order to identify possible explanations for our results, we searched the available literature on the toxicity of these two compounds (single or in mixture) due to their kinetics, absorption, and mechanisms of action. To date, to the best of our knowledge, the availability of studies evaluating the impact of simultaneous exposure to DEHP and Cd mixture on human and animal health is very limited, and currently, no studies on the reproductive area are available. In previous studies, detrimental effects of co-exposure to mixtures of contaminants belonging to the same class (heavy metals [83,84]; phthalate [85,86]) on reproductive functions were reported, mainly observing additive effects. A study in the mouse [63] reported the effects of vivo exposure to Mono-(2-ethylhexyl) phthalate (MEHP—the primary and the most toxic metabolite of DEHP) and Cd, individually and in their binary mixture, on target organs, such as liver, spleen, lungs, and kidneys. The results of this study are in line with ours, as they demonstrated that the toxic effects of these two environmental contaminants, both in single and in mixture, resulted in the suppression of cell proliferation, destruction of structure, integrity, permeability, and fluidity of cell membranes with increasing cell susceptibility to oxidative stress (OS). Joint toxic actions of these two compounds were hardly observed in this study. Another very recent study conducted on the insect Spodoptera littoralis revealed synergistic effects of Cd in combination with DEHP on the development, mass, and survival of adult subjects [87]. Thus, different species-specific, organ-specific, and cell-specific effects could be observed.

It can be hypothesized that OS observed on CCs exposed to the DEHP/Cd mixture in our study could be due to a direct increase in intracellular ROS linked to the ability of both contaminants to exert their toxic effect by increasing ROS levels.

At present, to the best of our knowledge, information on the absorption, kinetics, and mechanisms of action of DEHP/Cd mixtures in the COC of mammalian females is not reported. Information is available on the two individual compounds and, mainly, in other cell systems (Cd: [58,88]; DEHP: [38,43,89]) and on phthalates/Cd mixtures [63,87]. Thus, the way in which the mixture and/or the two compounds may have acted on the sheep COC can only be hypothesized in light of their known mechanisms of entry and action in other cellular systems. For Cd, cell entry is mediated by “ionic mimicry” by binding at sites for other essential/physiological bivalent cations [64], complexation with organic molecules (such as metallothionein and zinc-binding proteins, the antioxidant cysteine-containing tripeptide GSH, transferrin, cysteine, apolipoprotein A1, β2-microglobulin, albumin, lipocalin-2, immunoglobulin G, and Solute carrier SLC4A1 [90] and induction of OS at multiple cellular components have been reported [64]. Similar modes of entry and action have also been described in ovarian granulosa cells [64,91]. For DEHP, its entry in cells through plasma membranes and binding to peroxisome-specific nuclear proliferator-activated receptors (PPARs), also expressed in granulosa and theca cells, is known [38], as well as binding to estrogen receptors (ER) and androgen receptors (AR) [43,89], leading to OS. Based on these observations, the additive effect of the mix observed in the present study could be caused by the fact that the two compounds have different mechanisms of entry into the cell and mechanisms of action.

At the same time, we can hypothesize that the OS damage observed at the CCs level reflects a decline in the functional quality of the oocyte [92]. Reductions in mitochondria activity, ROS levels, and co-localization observed in oocytes could be interpreted as an effect of CCs, which, in defense of the oocyte, reduced these cellular activities. Indeed, CCs are mainly responsible for the oocyte OS defense by lowering the level of intracellular ROS production since the oocyte does not have the capacity on its own to mobilize all necessary antioxidant defense mechanisms [92,93,94]. It is known that CC gene expression and biochemical activity are directly influenced by oocyte conditions, as this protection is mediated through the production of antioxidant molecules such as glutathione and melatonin, along with the expression of several antioxidant enzymes. These enzymes play an even more critical role at the MII + PB stage, when the oocyte is transcriptionally silent, so these enzymes are supplied by the CCs instead of being expressed by the gametes [93,94,95]. This is supported by the observation that denuded oocytes were highly susceptible to OS [93,96,97]. CCs also protect the oocyte against OS during fertilization [98]. It has been demonstrated that CCs can specifically protect the oocyte from DEHP- [96,97] and Cd-induced [64] OS damage.

Therefore, considering that OS is the most relevant effect elicited by Cd and DEHP, it could be assumed that the mixture exerts a stronger effect on the COC than the individual compounds. The most plausible hypothesis is that, during the 24-hour IVM exposure, these compounds first enter the CCs, inducing OS, and are then transported into the oocyte through the cytoplasmic protrusions of the cumulus/corona radiata cells, altering its mitochondrial distribution and bioenergetic status.

Overall, it comes out that CC appears to be sensitive to mixture-induced OS due to its intrinsic morpho-functional nature of COC defensive barrier against chemical, physical, and biological pathogen agents, being made up of a series of concentric layers of CCs surrounding the oocyte. So, CCs can be considered sentinel cells of COC toxicity induced by environmental contamination.

Therefore, we could consider the potential role of CCs as biomarkers of DEHP/Cd mixture-induced COC damage. Research on toxicity biomarkers is fundamental in reproductive medicine because it can allow, within a medically assisted procreation cycle, for the performance of rapid, biochemical, or molecular analyses and the application of in vitro detoxifying therapies, improving the chances of obtaining fertilization, embryonic development and, therefore, pregnancies [92]. Moreover, in this context, this research contributes to exploring the potential in vivo applicability of identifying CCs as biomarkers with prognostic value to non-invasively assess oocyte quality and select it. Indeed, the physiological and morphological quality of CCs reflects the overall health of the oocyte, and they are usually partially or totally discarded during oocyte preparation for IVF/ICSI or can be collected in a non-invasive manner in biopsies consisting of small groups of cells, leaving unchanged the vitality of the COC, preserving its clinical use [99,100]. In recent years, the advent of “omics” has been considered [77,101,102] for these purposes. However, these approaches are complex and currently still related to the research area, so OS analysis on CCs at cytoplasmic [38] or DNA level [38,103,104] is relevant to opening new possibilities of optimization of clinical approaches to test oocyte quality before IVF or ICSI [92].

In conclusion, the results of the present study allow us to conclude that the toxicity of the examined DEHP/Cd mixture on the COC was stronger than that of individual contaminants, as demonstrated by OS at CC level and altered oocyte mitochondria pattern. To the best of our knowledge, this is the first study investigating the effects of the DEHP/Cd mixture on oocytes in a mammalian model. Cumulus cells appear sensitive to mixture-induced oxidative damage, suggesting their protective role toward the oocyte and their possible role as biomarkers of DEHP/Cd mixture-induced COC damage. Overall, these data suggest that these chemicals may interfere with ovarian function. Further studies are needed to identify possible protection mechanisms of the CCs from the action of these contaminants on the oocyte and to identify in vitro therapeutic strategies.

## 4. Materials and Methods

### 4.1. Chemicals

All chemicals for in vitro cultures and analyses were purchased from Sigma-Aldrich (Milan, Italy) unless otherwise indicated. For in vitro culture, DEHP was diluted in methanol (MeOH) at a concentration of 12.8 mM. The stock solution was stored at +4 °C until the day of use. A 100 mM stock solution of Cd chloride (CdCl_2_) was prepared by dissolving CdCl_2_ in distilled water. The stock solution was stored at room temperature until the day of use.

### 4.2. Collection of Ovaries

Ovaries were recovered at a local slaughterhouse (Siciliani s.r.l.; Palo del Colle, Bari, Italy) from juvenile ewes (less than 6 months of age) subjected to routine veterinary inspection in accordance with the specific health requirements stated. Ovaries were transported to the laboratory at room temperature within 4 h from slaughter.

### 4.3. COC Retrieval and Selection

Ovaries were processed by the slicing procedure for immature COC retrieval [64]. Follicular contents were released in sterile Petri dishes containing phosphate-buffered saline (PBS). Only undamaged COCs displaying oocytes with homogeneous cytoplasm and surrounded by at least three intact cumulus cells (CCs) layers were selected for in vitro culture under a Nikon SMZ18 stereomicroscope equipped with a transparent heating stage set up at 37 °C (Okolab S.r.l., Napoli, Italy) [105].

### 4.4. In Vitro Maturation (IVM)

COCs were in vitro cultured, as previously reported [106]. Briefly, in vitro maturation (IVM) medium was prepared from TCM199 with Earle’s salts, buffered with 5.87 mM 4-(2-hydroxyethyl)-1-piperazine-ethanesulfonic acid (HEPES) and 33.09 mM sodium bicarbonate, and supplemented with 200 mM L-glutamine solution, 2.27 mM sodium pyruvate, 2.92 mM calcium-L-lactate pentahydrate (1.62 mM Ca^2+^, 3.9 mM Lactate), 50 μg/mL gentamicin, 20% (*v*/*v*) fetal calf serum (FCS), gonadotropins (10 μg/mL of porcine follicle stimulating hormone, and luteinizing hormone (FSH/LH; Pluset^®^, Calier, Barcelona, Spain) and 1 μg/mL 17β estradiol. The medium was pre-equilibrated for 1 hour under 5% CO_2_ in an air atmosphere at 38.5 °C, then transferred (400 µL/well) in a 4-well dish (Nunc Intermed, Roskilde, Denmark) and covered with pre-equilibrated lightweight paraffin oil. In each experiment, approximately 20–25 COCs per condition were placed in a well of the four-well dish. IVM culture was performed for 24 h at 38.5 °C under 5% CO_2_ in air. During culture, COCs were alternatively exposed to 0.1 µM or 0.5 µM DEHP or a mixture with 0.1 µM CdCl_2_ and 0.5 µM DEHP/0.1 µM in IVM medium. Medium with 0.0005% MeOH, as DEHP vehicle was used as control. The working solutions were prepared by diluting the respective stock solutions in an IVM medium on the day of the experiment [38,44,64].

### 4.5. Cumulus Cell Isolation and Collection

After the IVM culture, CCs were removed from COCs by the denuding procedure. In detail, CCs were mechanically stripped from oocytes under stereomicroscopy using Gilson micropipettes (Middleton, WI, USA). COCs were gently pipetted up and down in TCM199 with 20% FCS and 80 IU hyaluronidase/mL. Denuded oocytes were collected and processed for further evaluations, as described below. Isolated CCs were collected in 0.5 mL RNAse-free tubes and washed 3 times in TCM199 with 20% FCS by centrifugation at 300× *g* for 2 min.

### 4.6. Oocyte and CC Staining for Mitochondrial and Intracellular ROS

In order to localize ooplasmic mitochondria and reactive oxygen species (ROS), ovine oocytes and CCs underwent a staining procedure with MitoTracker Orange CMTMRos (Thermo Fisher Scientific, Waltham, MA, USA) and H_2_DCF-DA as previously described [106,107]. Oocytes and CCs were washed three times in PBS with 3% BSA. CCs were centrifuged at 300× *g* for 2 min each time. After washing, oocytes and CC pellets were incubated for 30 min in the same medium containing 280 nM MitoTracker Orange CMTMRos at 38.5 °C under 5% CO_2_. The cells were gently shaken a few times during the incubation. Following incubation with MitoTracker, oocytes and CCs were washed thrice in PBS with 0.3% BSA and incubated for 15 min, at 38.5 °C under 5% CO_2_, in the same medium, containing 10 µM 2′,7′-dichlorodihydrofluorescein diacetate (H_2_DCF-DA) to detect the dichlorofluorescein (DCF) and localize intracellular sources of ROS. Oocytes and CC pellets were then washed in PBS without BSA and fixed overnight at 4 °C with 2% paraformaldehyde (PFA) solution in PBS [38]. Particular attention was paid to avoiding sample exposure to the light during staining and fixing procedures to reduce photobleaching. After fixation, CCs were centrifuged, and their pellets were resuspended in 7 μL PFA solution and subsequently added to a glass slide and covered with a cover glass that was sealed with nail polish. Microscope slides were kept at 4 °C in the dark until observation.

### 4.7. Oocyte Nuclear Chromatin Evaluation

Oocyte nuclear chromatin configuration was evaluated after fixation by oocyte staining with 2.5 μg/mL Hoechst 33258 in 3:1 (*v*/*v*) glycerol/PBS and mounting on microscope slides maintained at 4 °C in the dark until observation. Slides were examined under an epifluorescence microscope (Nikon Eclipse 600; ×400 magnification, Nikon Instrument, Firenze, Italy) equipped with a B-2A (346 nm excitation/ 460 nm emission) filter. Oocytes were evaluated in relation to their meiotic stage and classified as germinal vesicle (GV), metaphase to telophase I (MI to TI), MII with the 1st polar body (PB) extruded, or as degenerated for either multipolar meiotic spindle, irregular chromatin clumps, or absence of chromatin [106].

### 4.8. Assessment of Oocyte Mitochondria Distribution Pattern and Intracellular ROS 

Mitochondria localization in MII Oocytes was observed using a Nikon C1/TE2000-U laser-scanning confocal microscope (Nikon Instruments, Firenze, Italy) at ×600 magnification in oil immersion. A 543 nm helium/neon laser and G-2A filter (551 nm excitation and 576 nm emission) were used to point out the MitoTracker Orange CMTMRos. A 488 nm argon ion laser and the B-2A filter (495 nm excitation and 519 nm emission) were used to detect the DCF. To perform a 3D mitochondrial and ROS distribution analysis, oocytes were observed in 25 optical sections from the top to the bottom with a step size of 0.45 μm. The mitochondrial distribution pattern was evaluated based on previously reported criteria [106]. Thus, (a) perinuclear and subplasmalemmal distribution (P/S, with mitochondria more concentrated in the oocyte hemisphere, where the meiotic spindle is located and forming large granules in the cortical region) was considered as characteristic of healthy cytoplasmic condition; (b) homogeneous distribution (with small mitochondria aggregates throughout the cytoplasm) was considered as an indication of low energy cytoplasmic condition; (c) irregular distribution of mitochondria forming large mitochondrial clusters was considered as abnormal distribution. Concerning intracellular ROS localization, healthy oocytes were considered those with intracellular ROS distributed throughout the cytoplasm, together with areas/sites of mitochondria/ROS overlapping.

### 4.9. Quantification of Oocyte and CC Mitochondrial Membrane Potential (ΔΨm), Intracellular ROS Levels, and Mitochondria-ROS Co-Localization

In each individual MII oocyte and CCs, MitoTracker and DCF fluorescence intensities and overlap coefficient were measured using the EZ-C1 Gold Version 3.70 image analysis software platform for the Nikon C1 confocal microscope. The quantification analysis of the oocyte was performed at its equatorial plane. For CC analysis, ten fields of groups of cells per condition per trial were analyzed. The analysis was performed by drawing a circle area to select and analyze only oocyte and CC regions, including cell cytoplasm. The fluorescence intensity encountered within the programmed scan area (512 × 512 pixels) was recorded, and 16-bit images were obtained. Mitochondrial membrane potential and intracellular ROS levels were recorded as the fluorescence intensity emitted by MitoTracker and DCF probe, respectively, and expressed in arbitrary densitometric units (ADUs). Variables related to fluorescence intensity, such as laser energy, signal detection (gain), and pinhole size values, were held constant for all measurements. The degree of mitochondria/ROS co-localization was recorded as an overlap coefficient, indicating the overlap degree between MitoTracker Orange CMTMRos and DCF fluorescence signals. For mitochondria-ROS co-localization analysis, threshold levels were kept constant at 10% of the maximum pixel intensity.

### 4.10. Statistical Analysis

The proportions of oocytes showing the different chromatin configurations and mitochondria distribution patterns were compared among groups by chi-square test without or with Yates’ correction for contingency tables with small cell counts. The fluorescence intensity data of the MitoTracker CMTM Ros and DCF for quantitative analysis of the activity mitochondrial and intracellular ROS levels, respectively, and the co-localization mitochondria/ROS (overlap coefficient) data were compared among groups by one-way ANOVA test followed by Tukey’s post hoc test (GraphPad software 5.03, San Diego, CA). For each purpose, at least three replicates were performed. In each replicate, at least 100 COCs were processed, with a mean number of about 25 COCs per each exposure condition. Data (mean ± standard deviations) were transformed into percentages with respect to vehicle CTRL of the same replicate. Differences with *p* < 0.05 were considered to be statistically significant.

## Figures and Tables

**Figure 1 ijms-26-00005-f001:**
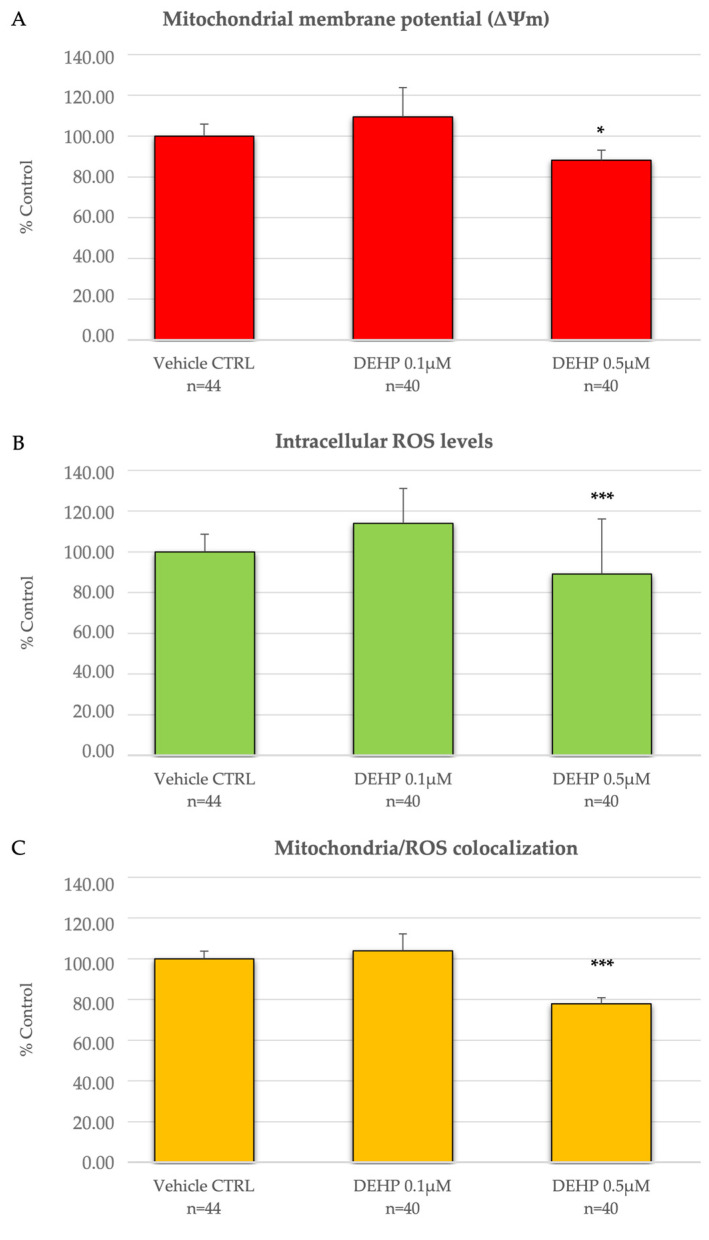
Dose-dependence curve of the in vitro effects of DEHP on mitochondrial membrane potential (ΔΨm), intracellular ROS levels, and mitochondrial/ROS co-localization in single metaphase II stage oocytes expressed as MitoTracker Orange CMTM Ros (**A**) and DCF (**B**) fluorescence intensities and overlap coefficients of MitoTracker Orange CMTM Ros and DCF fluorescent labeling (**C**). Values are means ± standard deviations and are expressed as percentages of vehicle CTRL. Numbers of analyzed oocytes per group are indicated on the bottom of each histogram. One-way ANOVA test followed by Tukey’s post hoc test, comparisons among groups; * *p* < 0.05 and *** *p* < 0.001 for DEHP-exposed vs. vehicle CTRL.

**Figure 2 ijms-26-00005-f002:**
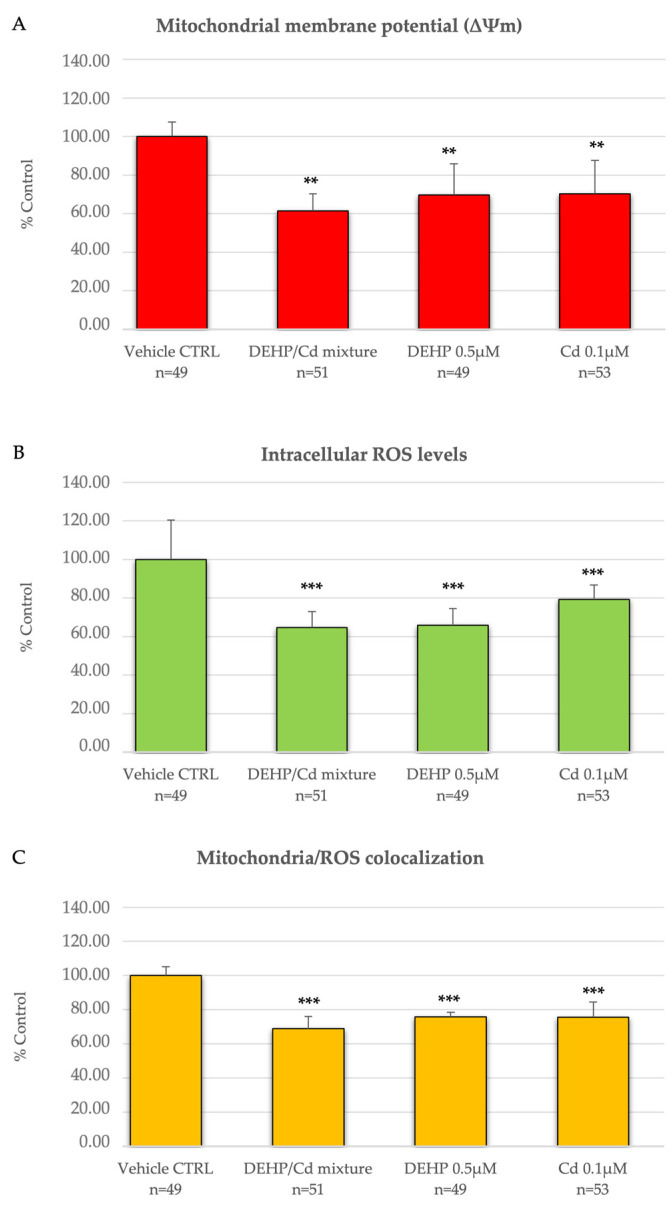
Effects of DEHP/Cd mixture and individual compounds on mitochondrial membrane potential (ΔΨm), intracellular ROS levels, and mitochondrial/ROS co-localization in single metaphase II stage oocytes expressed as MitoTracker Orange CMTM Ros (**A**) and DCF (**B**) fluorescence intensities and overlap coefficients of MitoTracker Orange CMTM Ros and DCF fluorescent labeling (**C**). Values are means ± standard deviations of examined oocytes and are presented as percentages of vehicle CTRL. Numbers of analyzed oocytes per experimental condition are indicated at the bottom of each bar. One-way ANOVA test followed by Tukey’s post hoc test, different superscripts indicate statistically significant differences: comparisons among groups; ** *p* < 0.01 and *** *p* < 0.001 for exposed vs. vehicle CTRL.

**Figure 3 ijms-26-00005-f003:**
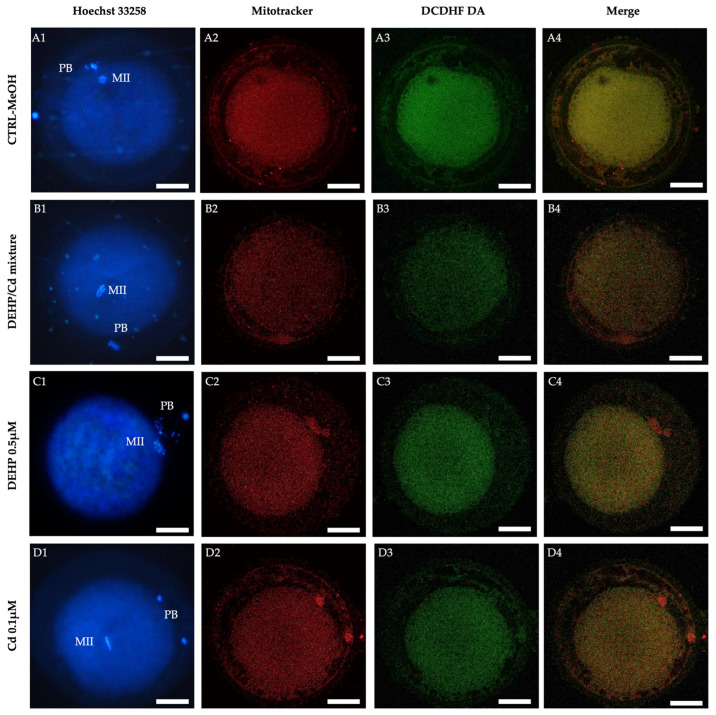
Mitochondrial distribution pattern and ROS localization in sheep matured oocytes exposed to DEHP/Cd mixture, DEHP and Cd. For each oocyte, corresponding UV light (blue pictures in (**A1**),(**B1**),(**C1**),(**D1**)) and confocal laser scanning images showing mitochondrial distribution pattern (red pictures in (**A2**),(**B2**),(**C2**),(**D2**)), intracellular ROS localization (green pictures in (**A3**),(**B3**),(**C3**),(**D3**)) and mitochondria/ROS merge (orange pictures in (**A4**),(**B4**),(**C4**),(**D4**)) are shown. Oocytes are representative of heterogeneous (perinuclear/subplasmalemmal; (**A**) and homogeneous (**B**–**D**) mitochondrial distribution patterns, respectively. White scale bar represents 40 µm.

**Figure 4 ijms-26-00005-f004:**
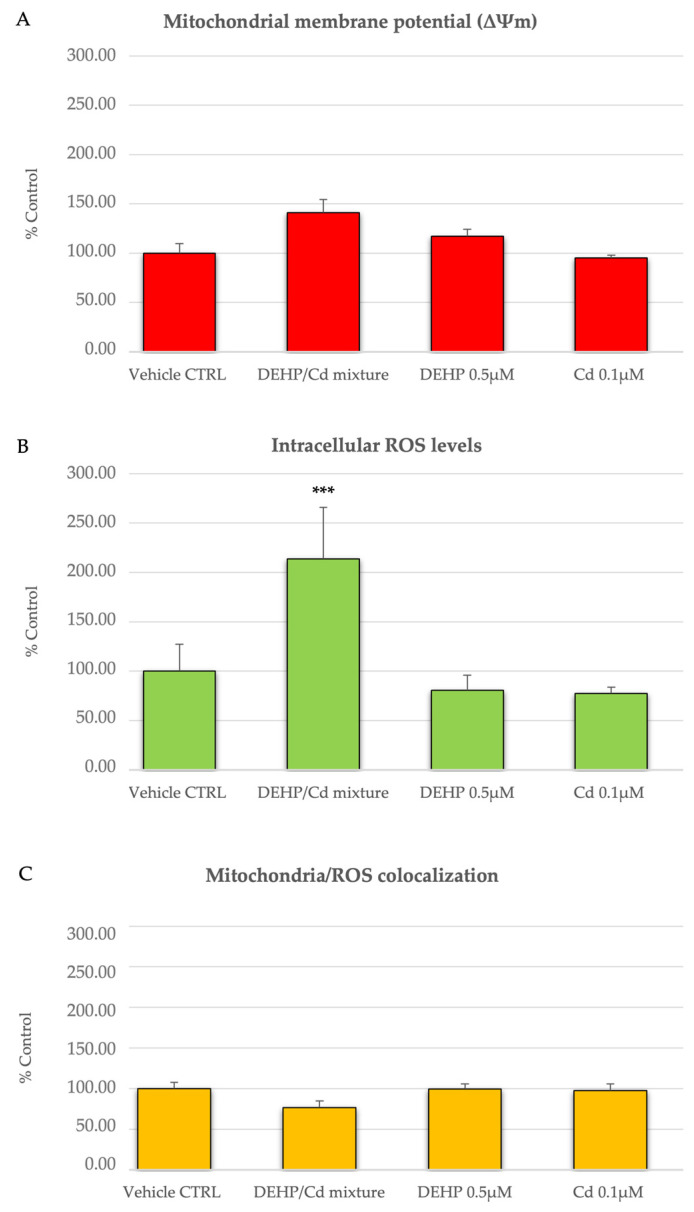
Effects of DEHP/Cd mixture on mitochondrial membrane potential (ΔΨm), intracellular ROS levels, and mitochondrial/ROS co-localization in CCs from sheep COCs expressed as MitoTracker Orange CMTM Ros (panel (**A**)) and DCF (panel (**B**)) fluorescence intensities and overlap coefficient of mitochondria/ROS co-localization (panel (**C**)). Values are means ± standard deviations of examined fields and are expressed as percentages of vehicle CTRL. Around 200 CCs per experimental condition were analyzed by LSCM. One-way ANOVA test followed by Tukey’s post hoc test: comparisons among groups; different superscripts indicate statistically significant differences: *** *p* < 0.001 for DEHP/Cd mixture-exposed vs. vehicle CTRL, 0.5 µM DEHP, and 0.1 µM Cd.

**Figure 5 ijms-26-00005-f005:**
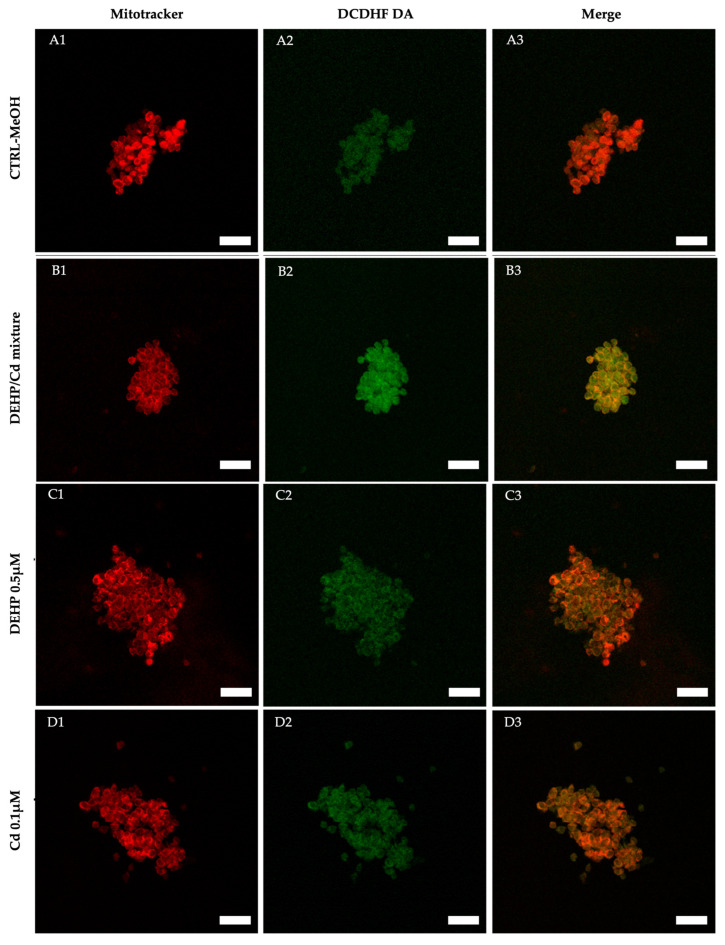
Photomicrographs of representative CCs from COCs of juvenile ewes matured in vitro in presence/absence of DEHP and Cd. Lanes show representative CC fields taken by laser scanning confocal microscopy. In each field, all CCs included in the yellow defined boundary underwent quantification analysis of mitochondrial membrane potential, ROS levels, and mt/ROS co-localization, whose results are presented in Figure 4. In columns 1 and 2, cells stained with MitoTracker Orange and DCF are shown, respectively, whereas column 3 shows the mitochondrial/ROS merge. Increased intracellular ROS levels (expressed as DCF fluorescent intensity) can be seen in the CCs exposed to DEHP/Cd mixture (**B2**) compared with the controls (**A2**). White scale bars represent 40 µm.

**Table 1 ijms-26-00005-t001:** In vitro effects of DEHP on oocyte nuclear chromatin configuration.

DEHP (µM)	Total Oocyte Number	Oocyte Number (%)			
		Germinal Vesicle	Metaphase I to Telophase I	Metaphase II and 1st Polar Body	Abnormal
**0 (CTRL)**	78	9 (11)	2 (3)	53 (68)	14 (18)
**0 (vehicle CTRL)**	72	6 (8)	5 (7)	44 (61)	17 (24)
**0.1**	75	2 (3)	6 (8)	40 (53)	27 (36)
**0.5**	76	9 (12)	10 (13)	40 (53)	17 (22)

Chi-square Test: NS. Data were obtained from three independent replicates.

**Table 2 ijms-26-00005-t002:** In vitro effects of DEHP on oocyte mitochondrial distribution pattern.

DEHP (µM)	Number of Oocytes Found at the MII Stage and Evaluated	Oocyte Number (%)		
		Perinuclear and Subplasmalemmal	Small Aggregates	Abnormal
**0 (CTRL)**	53	27 (51)	26 (49)	0 (0)
**0 (vehicle CTRL)**	44	21 (48) ^a^	23 (52) ^a^	0 (0)
**0.1**	40	10 (25) ^#^	27 (68)	3 (7)
**0.5**	40	8 (20) ^b^	32 (80) ^b^	0 (0)

Chi-square test with Yates’ correction: within each column, different superscripts indicate statistically significant differences: ^a, b^
*p* < 0.05; ^#^
*p* = 0.0537 not quite statistically significant. Data were obtained from three independent replicates.

**Table 3 ijms-26-00005-t003:** In vitro effects of DEHP/Cd mixture and individual compounds on oocyte meiotic maturation.

Condition	Total Number of Evaluated Oocytes	Oocytes Number (%)			
		Germinal Vesicle	Metaphase I to Telophase I	Metaphase II and 1st Polar Body	Abnormal
**0 (vehicle CTRL)**	121	11 (9)	12 (10)	73 (60)	25 (21)
**DEHP/Cd mixture**	120	9 (7)	13 (11)	73 (61)	25 (21)
**DEHP 0.5µM**	124	11 (9)	17 (14)	64 (52)	32 (26)
**Cd 0.1µM**	119	13 (11)	8 (7)	76 (64)	22 (18)

Chi-square Test: NS. Data were obtained from five independent replicates.

**Table 4 ijms-26-00005-t004:** In vitro effects of DEHP/Cd mixture and individual compounds on oocyte mitochondrial distribution pattern.

Condition	Number of MII Evaluated Oocytes	Oocyte Number (%)		
		Perinuclear and Subplasmalemmal	Small Aggregates	Abnormal
**0 (vehicle CTRL)**	49	22 (45) ^a^	27 (55) ^a^	0 (0)
**DEHP/Cd mixture**	51	5 (10) ^d^	42 (82) ^c^	4 (8)
**DEHP 0.5µM**	49	11 (22) ^b^	38 (78) ^b^	0 (0)
**Cd 0.1µM**	53	12 (23) ^b^	41 (77) ^b^	0 (0)

Chi-square with Yates’ correction: within each column, different superscripts indicate statistically significant differences: ^a, b^
*p* < 0.05; ^a, c^
*p* < 0.01; and ^a, d^
*p* < 0.001. Data were obtained from four independent replicates.

## Data Availability

The original contributions presented in this study are included in the article. Further inquiries can be directed to the corresponding author.

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
