# Peer review of "In Vitro Toxicity of a DEHP and Cadmium Mixture on Sheep Cumulus–Oocyte Complexes"

_ijms, 2024, doi:10.3390/ijms26010005_

Round 1

Reviewer 1 Report

Comments and Suggestions for Authors

This study analyzed the effects of a DEHP/Cd mixture on nuclear and cytoplasmic maturation of sheep cumulus-oocyte complexes compared with single compounds. They found that oocytes suffered the same damage when exposed to the mixture or separate compounds whereas CCs displayed oxidative stress only upon mixture exposure. They provided new insights in  toxicity of a DEHP and Cadmium in reproduction. However, these issues should be cared.
1, how many technological repeats were did in this study? it is very important for this study, such as in table 1 and 2, showed there was no technological and biological repeats.
2,  in vitro fertilization is the most direct method to assessing the quality of oocytes. Therefore, suggest adding in vitro fertilization experiment, if possible.

3,in figure 1, 2 and 4, the standard deviation (bar) looks too great, which indicates the results not stable. Therefore, it is recommended to recalculate.

4, in figure 3 and 5, MII is a stage of oocyte meiotic maturation, not the nuclear of oocyte. Moreover, fluorescence staining is only a qualitative analysis, it is recommended to conduct a qualitative analysis on the grayscale values.

Comments on the Quality of English Language

Language and Grammar: Careful proofreading and editing are still required to address some remaining typos and grammatical issues.

Author Response

This study analyzed the effects of a DEHP/Cd mixture on nuclear and cytoplasmic maturation of sheep cumulus-oocyte complexes compared with single compounds. They found that oocytes suffered the same damage when exposed to the mixture or separate compounds whereas CCs displayed oxidative stress only upon mixture exposure. They provided new insights in toxicity of a DEHP and Cadmium in reproduction. However, these issues should be cared.

1, how many technological repeats were did in this study? it is very important for this study, such as in table 1 and 2, showed there was no technological and biological repeats.

In this study, the number of replicates was reported in the result sections. However, following the reviewer's suggestion, a sentence was added in methods (Statistical analysis) to indicate that all experimental tests were performed at least in three replicates. Numbers of replicates have been added in each table. Numbers of oocytes analyzed for each experimental condition are also reported, i.e., at the base of each histogram and in tables.

2,  in vitro fertilization is the most direct method to assessing the quality of oocytes. Therefore, suggest adding in vitro fertilization experiment, if possible.

The reviewer is in right in reporting that in vitro fertilization (IVF) is the most direct method for assessing oocyte quality since it simultaneously expresses nuclear and cytoplasmic oocyte quality. However, this technique may add variables determined by sperm quality that can affect the results. Consequently, in this first part of the study on mixture effects, we decided to focus our attention only on the oocyte and the bioenergetic analysis was performed as measurement of its cytoplasmic quality. Indeed, other than oocyte nuclear maturation, analyzed by chromatin configuration, this study investigated oocyte bioenergetics, both on qualitative aspects, relating to the distribution pattern of active mitochondria, and quantitative ones, relating to the intensity of fluorescence probe signals: the MitoTracker, measuring mitochondria membrane potential, the DFC, measuring intracellular levels of reactive oxygen species and their overlap (colocalization) indicative of healthy cell status.Future studies will focus on the effects of Cd/DEHP mixture on oocyte fertilization and embryo development by IVF and IVC.

3,in figure 1, 2 and 4, the standard deviation (bar) looks too great, which indicates the results not stable. Therefore, it is recommended to recalculate.

We thank the reviewer for giving us the opportunity to improve the quality of our data analysis and thus of the manuscript. As suggested, data were recalculated. For each oocyte, data of ADU and overlap coefficient were converted in percentage of vehicle CTRL of the same replicate. New mean ± standard deviations were obtained and are represented in new figures 1, 2 and 4. The results of statistical analysis were confirmed.

4, in figure 3 and 5, MII is a stage of oocyte meiotic maturation, not the nuclear of oocyte.

We apologize but we do not fully understand this comment. As far as we wanted to say, in Tables 1 and 3 and in Figure 3, oocyte nuclear maturation is meant as modifications of oocyte nuclear chromosomal compIement from the GVBD (Germinal Vesicle Breakdown) to the MII stage. Indeed, it is known that, in mammals, oocyte meiosis has a coordination between nuclear and cytoplasmic maturation and that “Nuclear maturation encompasses the processes reversing meiotic arrest at prophase I and driving the progression of meiosis to metaphase II whereas cytoplasmic maturation refers to the processes that prepare the egg for activation and preimplantation development.” (Eppig et al., 1996, Reprod Fertil Dev. 1996;8(4):485-9. doi: 10.1071/rd9960485. “Coordination of nuclear and cytoplasmic oocyte maturation in eutherian mammals”).

Moreover, fluorescence staining is only a qualitative analysis, it is recommended to conduct a qualitative analysis on the grayscale values.

For oocytes, both qualitative (Table 2 and Figure 3) and quantitative (Figures 1 and 2) analysis of MitoTracker and DCF fluorescent signals were performed.

For CCs, representative images of the stained cells are shown in figure 5 and the results of the quantitative analysis are reported in figure 4.

Details on quantification analyses are reported in the materials and methods section: "4.9. Quantification of oocyte and CC mitochondrial membrane potential, intracellular ROS levels, and mitochondria-ROS co-localization.”

5, Language and Grammar: Careful proofreading and editing are still required to address some remaining typos and grammatical issues.

We thank the reviewer for giving us the opportunity to improve the quality of our manuscript. As suggested, the English language of the manuscript has been reviewed and revised by one of the authors, who is an expert with advanced proficiency in English.

Reviewer 2 Report

Comments and Suggestions for Authors

This manuscript presents that toxic effect of DEHP and Cd mixture on the two cell types of the female gamete was might variable depend on ovarian function.

Overall, the experimental method, results, and discussions are clearly organized, so it is considered a meaningful paper for researchers related to reproductive toxicology.

For publication, the following points should be revised.

- Discussion is needed on the differences in absorption and kinetics of single or mixed EDCs in response to exposure to COCs.

- Additional references showing that cumulus cells are more sensitive to oxidative damage caused by a mixture of DEHP and Cd and that the oocytes have protective functions are recommended.

- The potential of cumulus cells as a selective biomarker of COC damage needs to be supplemented with a discussion of the potential in vivo applicability

Author Response

This manuscript presents that toxic effect of DEHP and Cd mixture on the two cell types of the female gamete was might variable depend on ovarian function.

Overall, the experimental method, results, and discussions are clearly organized, so it is considered a meaningful paper for researchers related to reproductive toxicology.

For publication, the following points should be revised.

- Discussion is needed on the differences in absorption and kinetics of single or mixed EDCs in response to exposure to COCs.

- Additional references showing that cumulus cells are more sensitive to oxidative damage caused by a mixture of DEHP and Cd and that the oocytes have protective functions are recommended.

- The potential of cumulus cells as a selective biomarker of COC damage needs to be supplemented with a discussion of the potential in vivo applicability.

We thank the reviewer for positive comments on our manuscript. Following his/her suggestions, discussion on the raised points has been expanded. Most of cited references concern the individual compounds, considering that literature on DEHP and Cd mixtures is not widely available to date and this is the innovative aspect of the study.

Round 2

Reviewer 1 Report

Comments and Suggestions for Authors

All my comments have been addressed.